# Parameterized Synthetic Text Generation with SimpleStories

**Lennart Finke** *
ETH Zürich
lfinke@ethz.ch

**Chandan Sreedhara**
Independent

**Thomas Dooms**
University of Antwerp

**Mat Allen**
Dioptra

**Emerald Zhang**
UT Austin

**Juan Diego Rodriguez**
UT Austin

**Noa Nabeshima**
Independent

**Thomas Marshall**
EleutherAI

**Dan Braun**
Goodfire †

## Abstract

We present SimpleStories, a large synthetic story dataset in simple language, consisting of 2 million samples each in English and Japanese. Through parameterizing prompts at multiple levels of abstraction, we achieve control over story characteristics at scale, inducing syntactic and semantic diversity. Ablations on a newly trained model suite show improved sample efficiency and model interpretability compared to the TinyStories dataset. We open-source all constituent parts of model creation, hoping to enable novel ways to study the end-to-end training process. As a byproduct, we move the frontier regarding the fewest-parameter language model that outputs grammatical natural language. The dataset and code can be accessed at https://huggingface.co/datasets/SimpleStories/SimpleStories and https://github.com/simple-stories/simple_stories_generate.

## 1 Introduction

"Once upon a time, clever people made language models. They were very useful, but worked like a magic box, and were hard to understand. But then came a happy surprise — a big book of simple stories that helped the people make simple language models." Such a tale might be told about the TinyStories dataset of Eldan and Li [2023]. It has greatly aided the progress towards a mechanistic understanding of LLMs by creating small model organisms trained on synthetic children's stories. Its stated research objective is to distill the concepts of grammar and reasoning into a text corpus by abstracting away factual knowledge — an idea that remains highly relevant as part of a broader discussion in the context of data-constrained training [Villalobos et al., 2022].

However, we found that the TinyStories dataset has two critical issues. First, it is formulaic; to illustrate, 59% of stories contain the string 'Once upon a time' verbatim. Second, it is unlabeled, which hinders the application of supervised methods to it, i.e. finetuning on a subset of the data. It is additionally only available in English, not entirely open-source, and contains many encoding artifacts, duplications, as well as graphic descriptions of violence unsuitable for the children's story setting.

---

*Corresponding author.
†Work done at Apollo Research.

39th Conference on Neural Information Processing Systems (NeurIPS 2025) Track on Datasets and Benchmarks.

Here, we address these issues through the creation of a new dataset and model suite, in a comparatively cost-effective manner. Our main contributions are: (1) SimpleStories, a new fully open-source synthetic dataset consisting of simple yet diverse language suitable for pretraining and interpretability research, (2) detailed analysis of the diversity of SimpleStories, and (3) a suite of high-quality language models trained on SimpleStories, using a custom tokenizer.

Our datasets and trained models are openly available at `https://huggingface.co/SimpleStories`, with dataset generation code at `https://github.com/simple-stories/simple_stories_generate` and an interactive dataset visualization using text embeddings at `https://fi-le.net/simplestories`. This enables other researchers to either use our dataset or generate more samples of their own. Our training sets contain around 2 million samples per language, and the designated test sets each contain around 20 thousand samples. The training code is available at `https://github.com/simple-stories/simple_stories_train`. Unlike the original TinyStories work, we open-source our story generation and model training code to enable the community to create variations of the datasets and model architectures.

## 2    Methods

Like TinyStories, we generate our dataset using commercial LLMs with template prompts that elicit simple language (full prompt in Appendix A). We use GPT-4o-mini-2024-07-18, which offers improved capabilities and alignment compared to the GPT-3.5 and GPT-4 models used in TinyStories. We are faced with the challenge of lexical diversity in LLM-based text generation—where specific phrases and expressions remain overrepresented even at high sampling temperatures—and therefore constrain each story to begin with one particular part of speech (adjective, adverb, noun, or preposition) and initial letter, with letter frequencies drawn from a reference corpus. Content diversity and labeling can be solved together through the following procedure. Instead of prompting for a selection of common words that should be used in the completion, we specify a topic, an overarching theme or feel, a writing style and a narrative feature; for the complete list, see Appendix A. To allow the study of phenomena relevant to alignment, we take care to represent potentially useful concepts such as "Cooperation", "Betrayal" or "Long-Term Thinking". On a subset of samples, we additionally prompt for a specific grammar feature or ask the LLM to assume an archetypal author persona. These categories are applicable across many languages, but their content may not be, and thus, one should balance an international perspective with language-specific tropes and traditions. We generate multiple stories simultaneously with the same parameters, which can reduce costs by saving on input tokens and lead to variations over an underlying idea, i.e. different instantiations of the same story structure. Together with the introduction of entropy through initial parts of speech and letter constraints, this disambiguates generations from the first token onwards, even across millions of stories. We can therefore use Nucleus Sampling [Holtzman et al., 2019] with $p = 0.9$ at temperature 1, increasing adherence to the given constraints.

Focusing specifically on training interpretable language models, we anticipate demand for word-level tokenization prompt the model to use only a limited vocabulary. One challenge is that the generating model will typically use proper names, some of which are imaginary. We prompt against this by instructing to compose proper names from common words and to use names from a given list. Finally, looking for issues with unsuitable content, we found that better alignment of production language models has fortunately led to the absence of violent stories, for example. To help with filtering and fine-grained usage, we precompute relevant metrics such as word count and Flesch-Kincaid reading grade [Kincaid, 1975].

## 3    Results

To evaluate our dataset, we compare its syntactic, lexical and semantic diversity in English with TinyStories. In doing so, the methods used in the original TinyStories work are a natural choice.

### 3.1    Lexical Diversity

We compute the most common $n$-grams in a random 10% subsample of both datasets, and greedily filter a descending frequency-sorted list for $n$-grams which do not overlap with a previous $n$-gram on more than $n - 2$ words. Through this, we can not only find the most common phrases, but also

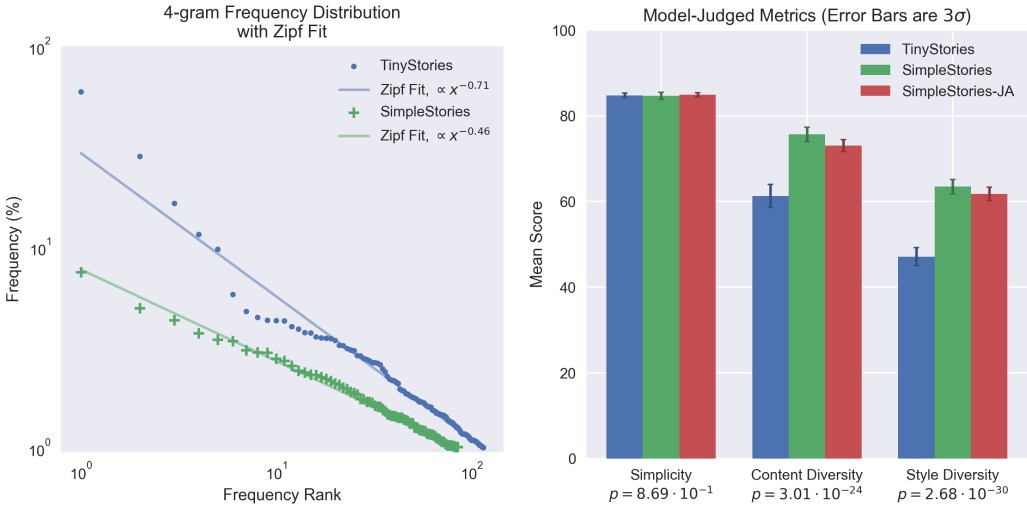

Figure 1: **Left**: an evaluation of lexical diversity through 4-gram frequencies by frequency rank, on a subsample of 10% of either dataset. **Right**: an evaluation of semantic diversity and simplicity through model-as-a-judge. The $3\sigma$ confidence intervals assume quantiles of a normal distribution with sample mean and variance. The uncorrected p-values each represent a Wilcoxon Rank-Sum Test with $N = 200$ and the null hypotheses that the score of either of the distributions have equal mean to the TinyStories mean.

analyze the distribution of their frequencies, i.e. the percentage that an $n$-gram occurs in one sample. They approximately follow a Zipf distribution, echoing empirical findings in large natural corpora [Ha et al., 2009]. As seen in Fig. 1, SimpleStories has a much more varied distribution of 4-grams. However, it has longer samples with an average and standard deviation of $224.6 \pm 103.8$ words as opposed to TinyStories' $175.4 \pm 80.2$ words. The highly common outliers in TinyStories stem from near-identical first sentences of different samples, which we avoided with the prompting procedure described in Section 2. Table 1 shows the most frequent 4-grams. Calculating the Flesch-Kincaid reading grade over the whole dataset, we find a mean grade and standard deviation of $3.08 \pm 1.24$ for SimpleStories as opposed to TinyStories' $2.55 \pm 1.49$. In total, our English dataset contains 452 million words, or 602 million GPT-2 tokens. For the Japanese dataset, we provide the most common 5-grams in Table 6, and confirm that no anomalous high-frequency phrases are present.

We also measure the compression ratio and Self-BLEU homogenization score, two diversity metrics with low mutual correlation [Shaib et al., 2025], on a random subsample of size 1000 for each dataset via the `diversity` package (https://github.com/cshaib/diversity). Computing the compression ratio allows us to measure diversity in terms of document compression relative to original size —higher compression ratios imply more redundancy. Self-homogenization scores capture aggregate similarity by computing the mean similarity over every pair of stories in the dataset. As seen in Fig. 3, SimpleStories is less repetitive than TinyStories, achieving significantly lower scores on both metrics.

Furthermore, we compute the *n-gram diversity score* (NGD) for n-gram sequences of length 1 through 10. [Shaib et al., 2025] represents n-gram diversity as the ratio of unique n-gram counts to all n-gram counts in a document, allowing us to capture repeated sequences in addition to single-token diversity. We observe from Fig. 3 that stories from SimpleStories result in higher NGD scores than those from TinyStories, particularly for larger values of n. SimpleStories contains stories with varied sequences and fewer repetitions, while the text from TinyStories reuses the same phrases repeatedly.

## 3.2 Semantic Diversity

We find inspiration in Eldan and Li [2023] by using a variant of their GPT-Eval (what has since been termed model-as-a-judge) to compare the semantic diversity of our dataset. We think of our text synthesis problem as constraint optimization — while keeping stories simple, produce as much variation in content and style as possible. We therefore instruct GPT-4o-mini to evaluate simplicity,

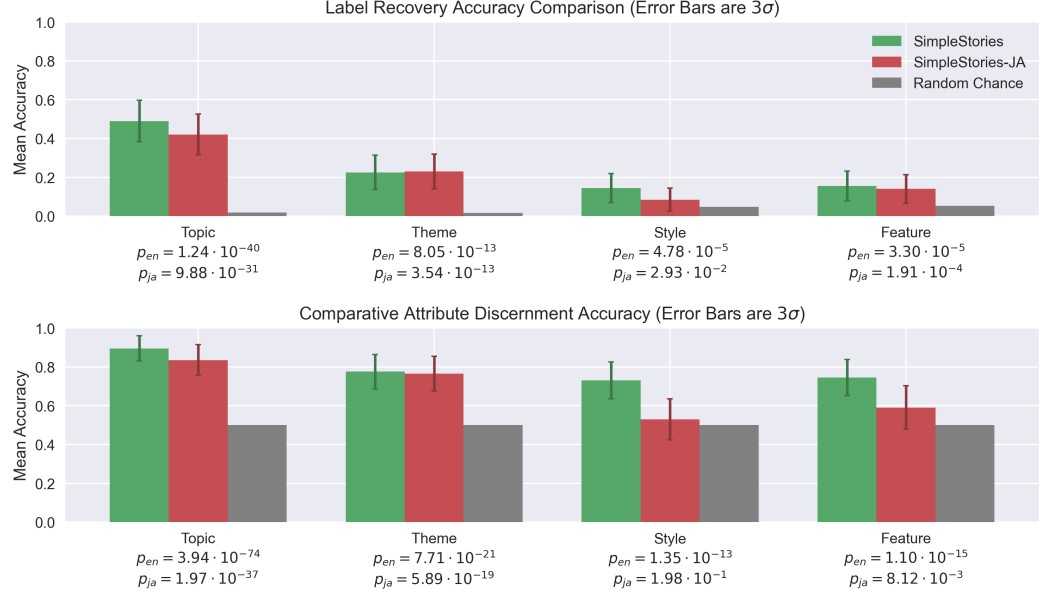

Figure 2: **Top**: Evaluation of label quality by accuracy of a judge model, GPT-4o-mini, annotating the story text versus the labels stemming from the generation process. The uncorrected p-values are from one-sided one-sample z-tests with $N = 200$ and the null hypothesis that the accuracy is no different from random guessing. **Bottom**: A judge model, o4-mini, distinguishing two pairs of stories, where one pair has two of the same label in the given category, versus a pair with two different labels. Again a z-test and $N = 200$ are used. This eliminates bias for specific labels inherent in the judge model.

content diversity and style diversity given 4 randomly drawn stories from the same dataset, on a scale from 0 to 100. Before this, we ask for a justification of the grading that does not enter our evaluation to induce the accuracy gains that typically result from chain-of-thought [Wei et al., 2022]. For exact prompts, see Appendix B. We perform a Wilcoxon Rank-Sum Test on the resulting scores with the null hypothesis that either the English or Japanese datasets are identical in the three metrics to the TinyStories dataset, with a total of $N = 200$ measurement units (therefore 800 stories). As seen in Fig. 1, the model-judged diversity for our datasets is much greater, with an insignificant model-judged difference in simplicity.

### 3.3 Syntactic Diversity

We evaluate syntactic diversity of our English dataset through the distribution of part of speech (POS) tag sequences using the `diversity` package. Shaib et al. [2024] defines templates as the most common POS sequences of length n in the corpus. We evaluate both *template rate* (the fraction of stories that contain at least one template) and *template-per-token* (the total number of templates in the entire corpus, normalized by word count). With $n = 6$ and considering the most frequent one hundred POS sequences, SimpleStories has a template rate of 88.9, as opposed to TinyStories' 100, and a template-per-token of 0.016, versus 0.026 for TinyStories. In other words, SimpleStories has greater syntactic diversity, with many stories containing less common POS sequences. A list of the most common POS sequences and their frequencies across both TinyStories and SimpleStories is shown in Table 5.

### 3.4 Labeling

Our method produces labels for each sampled story, but it is unclear a priori whether these convey meaningful information. Consequently, we test this by prompting a judge model (again GPT-4o-mini with chain-of-thought) to recover the labels present for every data point, given the list of all possible values in the dataset as answers to choose from. Using $N = 200$ samples per language and label category, we reject the null that this process is no better than random guessing for each label, at $\alpha = 0.001$. For the English dataset, the accuracy for recovering the topic, $0.49$, is particularly good,

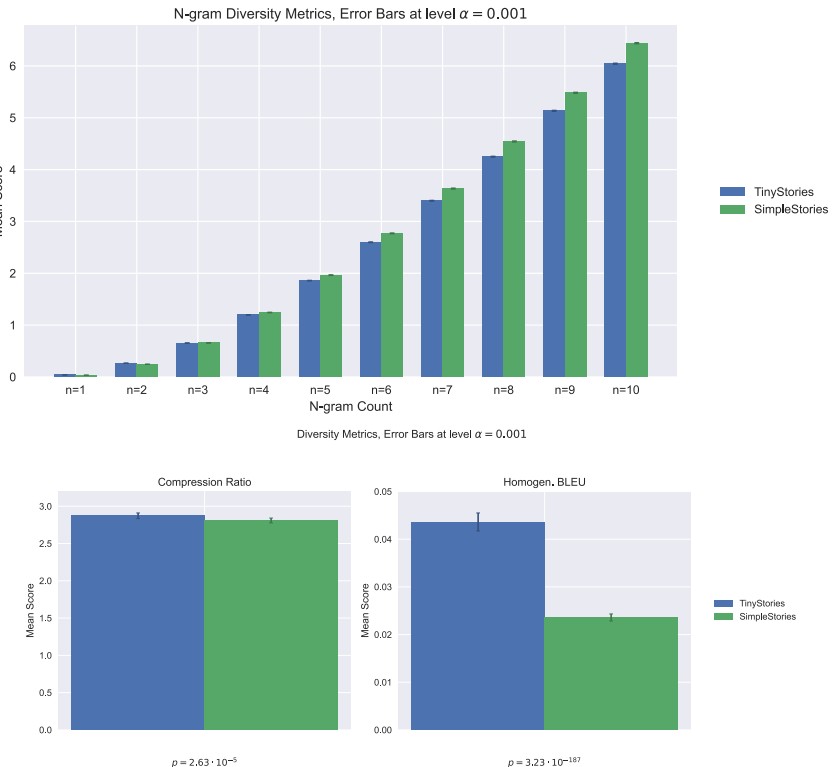

Figure 3: **Top**: N-gram diversity scores from TinyStories and SimpleStories Datasets, n-gram counts from 1 to 10. Higher n-gram diversity indicates less repetitive, more unique texts. **Left**: an evaluation of lexical diversity through compression ratio, on a size 1000 subsample of either dataset. **Right**: an evaluation of lexical diversity through Self-BLEU homogenization. The $3\sigma$ confidence intervals assume quantiles of a normal distribution with sample mean and variance. The uncorrected p-values represent the one-way ANOVA and the null hypotheses that the scores of both distributions have equal mean.

whereas the more abstract labels "theme", "style" and "narrative feature" are at $0.225, 0.145, 0.155$, as seen in Fig. 2.

Another experiment answers the same question of label quality, but eliminates a potential bias (or prior probability) of the judge model for a given label. We ask the judge (o4-mini) to discriminate between two pairs of stories, where one pair has two stories of the same label in a given category, while the other pair has two different labels. Using $N = 200$, we can detect much better than chance performance on all but the labels "style" and "feature" for the Japanese dataset. The highest accuracies were observed in the English dataset for labels "topic" and "theme", at $0.90$ and $0.78$, and in the Japanese dataset for the same labels at $0.84$ and $0.77$.

### 3.5 Training and Evaluation

We train multiple models with different parameter sizes and compare their performance against TinyStories. As detailed in Table 2, our SimpleStories models range from 1.25M to 35M parameters (including embedding parameters). We use AdamW [Kingma, 2014, Loshchilov and Hutter, 2017] with a learning rate of $10^{-4}$, a batch size of 128, gradient clipping, and a cosine learning rate decay with 100 steps of linear warmup. All models can be trained on a single A100 GPU in 12 hours.

While TinyStories uses the GPT-2 tokenizer with a vocabulary size of 50,257 tokens, we implement a custom WordPiece tokenizer [Schuster and Nakajima, 2012] with a significantly reduced vocabulary size of 4,096 tokens. Our tokenizer incorporates morphological analysis of the SimpleStories corpus to identify common English affixes (prefixes like "un", "re" and suffixes like "ed", "ing", "ly") which are then included as part of its initial alphabet.

Table 1: Top 20 most frequent 4-grams in TinyStories and SimpleStories Datasets, filtered to exclude overlaps of more than two words. Tokens are separated by a dot.

| TinyStories | | SimpleStories | |
|---|---|---|---|
| Frequency | Phrase | Frequency | Phrase |
| 59.38% | once upon a time | 7.49% | took a deep breath |
| 28.24% | there was a little | 4.95% | the sun began to |
| 16.52% | a little girl named | 4.32% | from that day on |
| 11.55% | a time there was | 3.71% | felt a spark of |
| 9.73% | from that day on | 3.46% | felt a rush of |
| 5.79% | a little boy named | 3.40% | as the sun set |
| 4.77% | to play in the | 3.06% | felt the weight of |
| 4.45% | was so happy and | 2.98% | with a deep breath |
| 4.32% | do you want to | 2.97% | a girl named mia |
| 4.30% | went to the park | 2.77% | thought for a moment |
| 4.28% | she loved to play | 2.71% | a boy named leo |
| 4.01% | to play with her | 2.56% | the end of the |
| 3.90% | the little girl was | 2.42% | was not just a |
| 3.75% | did not want to | 2.37% | the day of the |
| 3.73% | it was time to | 2.31% | felt a sense of |
| 3.57% | but it was too | 2.30% | laughter filled the air |
| 3.53% | there was a big | 2.25% | it was time to |
| 3.51% | was so happy that | 2.20% | a boy named samuel |
| 3.49% | there was a boy | 2.15% | at the edge of |
| 3.42% | they like to play | 2.09% | the magic of the |

Table 2: Architecture configurations of newly trained models. For the exact implementation, see the training repository code. Note that throughout this work, if not specified otherwise, the parameter counts in our model names include embedding parameters, whereas those in TinyStories model names do not include embedding parameters. We aim to achieve better performance at the same parameter count, even when including embedding parameters only for our models. Note that this discrepancy has led to misunderstandings in the past; see [Pearce and Song, 2024] for an in-depth discussion of one such case. We recommend future research in the search for the "smallest model that outputs grammatical English" to count all model parameters to avoid confusion.

| Model Name | n_layers | d_model | n_heads | d_vocab | n_params |
|---|---|---|---|---|---|
| SimpleStories-35M | 12 | 512 | 8 | 4096 | 35 million |
| SimpleStories-30M | 10 | 512 | 8 | 4096 | 30 million |
| SimpleStories-11M | 6 | 384 | 6 | 4096 | 11 million |
| SimpleStories-5M | 6 | 256 | 4 | 4096 | 5 million |
| SimpleStories-1.25M | 4 | 128 | 4 | 4096 | 1.25 million |
| TinyStories-33M | 4 | 768 | 12 | 50257 | 68 million |

Model generations are evaluated using a variant of the GPT-Eval framework, which scores outputs on four metrics: originality, coherence, grammar, and quality (1-100 scale). Figure 4 shows that our models consistently outperform TinyStories-33M in all metrics despite having considerably fewer parameters, with notable improvements in coherence and quality.

To isolate the impact of our tokenization strategy, architectural choices, and dataset quality, we performed a comprehensive ablation study. We evaluated TinyStories-33M with multiple configurations: GPT-2 tokenizer (arch-gpt2, tok-gpt2), Llama architecture with GPT-2 tokenizer (arch-llama, tok-gpt2), a custom 4096-token tokenizer optimized specifically for TinyStories (arch-llama, tok-custom-tinystories), and our custom 4096-token SimpleStories tokenizer with Llama architecture (arch-llama, tok-custom-simplestories). We also tested SimpleStories-35M with both GPT-2 and our custom SimpleStories tokenizer. As shown in Figure 5, using custom tokenizers consistently boosts performance compared to the GPT-2 tokenizer for both datasets. The TinyStories-optimized

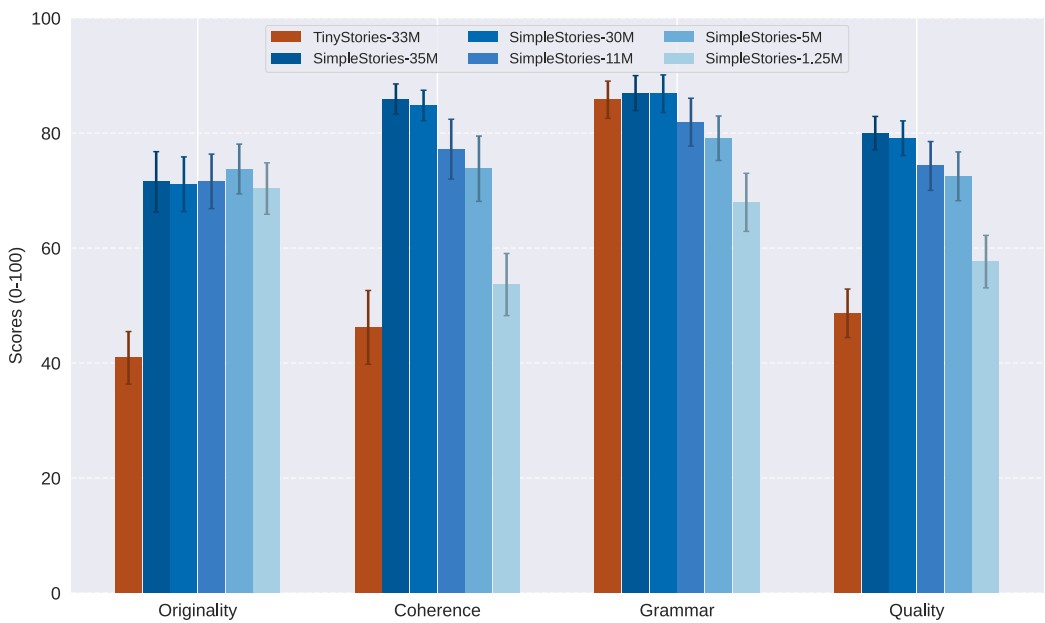

Figure 4: Performance comparison of language models across four evaluation metrics: originality, coherence, grammar, and quality. The visualization shows scores (0-100 scale) for six different models: TinyStories-33M and five SimpleStories variants with parameter counts ranging from 1.25M to 35M. Each model was evaluated on $N = 200$ generated stories using a model-as-a-judge with GPT-4o-mini. Error bars represent $3\sigma$ confidence intervals.

tokenizer provides meaningful improvements, and the Llama architecture further enhances performance, particularly in originality and coherence. However, even after optimizing TinyStories with its own custom tokenizer and improved architecture, all TinyStories variants remain substantially below SimpleStories-35M with its custom tokenizer across all metrics. This demonstrates that while tokenization strategy and architecture meaningfully contribute to model effectiveness, the diversity and linguistic richness of the dataset are the primary drivers of the performance improvements we observe.

Interestingly, the original TinyStories-33M model exhibits unexpectedly strong grammar performance, nearly matching our best model despite lower scores on other metrics. This suggests that the original TinyStories training approach may have specifically emphasized grammatical correctness. This asymmetry in performance across metrics underscores how evaluation metrics can reveal architectural biases and trade-offs that are not immediately evident from overall performance measures.

### 3.6 Probing

We measure the degree to which our models learn the labels through probes [Alain and Bengio, 2018]. This reflects their high-level understanding of the context, rather than simple token generation. We learn probes on a story level through a learned token-pooling operation followed by a linear head (see Appendix C for details). As shown in Figure 6, the probes outperform the judge model by a significant margin for all but the smallest model on an evaluation set. Probe accuracy is notably lower (by approximately 30%) near the initial layers of the larger models (30M and 35M parameters). This suggests the targeted labels are less explicitly represented in the output tokens, indicating that the model learned to detect these features internally. We stress-test this claim through ablations and fuzzing in Figure 8.

### 3.7 Interpretability

The goal of this synthetic dataset is to allow the creation of more interpretable model organisms. To study this, we compare differences in prominent patterns learned for two single-layer models trained

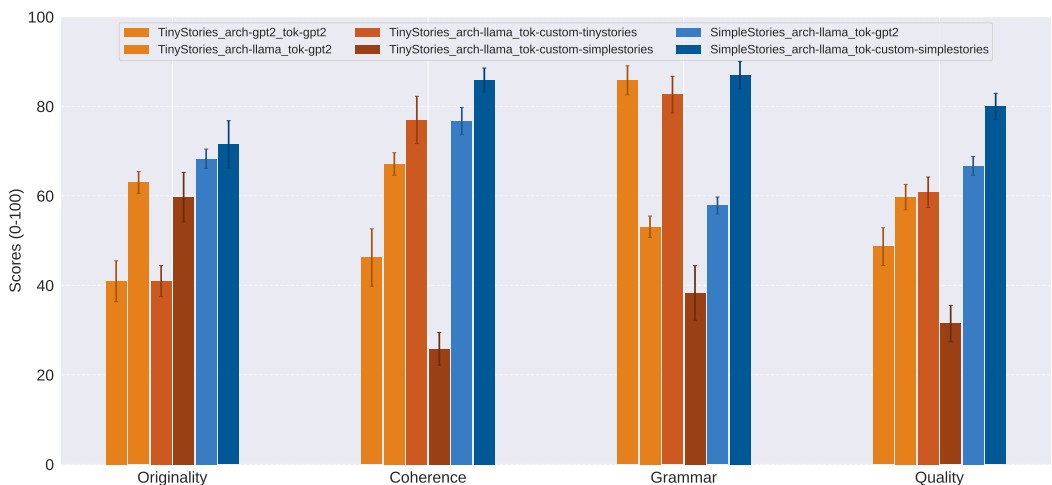

Figure 5: Ablation study comparing tokenizer, architecture, and dataset effects on four evaluation metrics. We evaluated: Dataset TinyStories vs SimpleStories; Architecture - GPT-2 (arch-gpt2) vs Llama (arch-llama); Tokenizer - GPT-2 (tok-gpt2) vs custom 4096-token tokenizers (tok-custom-tinystories and tok-custom-simplestories, trained on their respective datasets). Models evaluated on N = 200 generated stories using GPT-4o-mini as judge, with error bars showing $3\sigma$ confidence intervals.

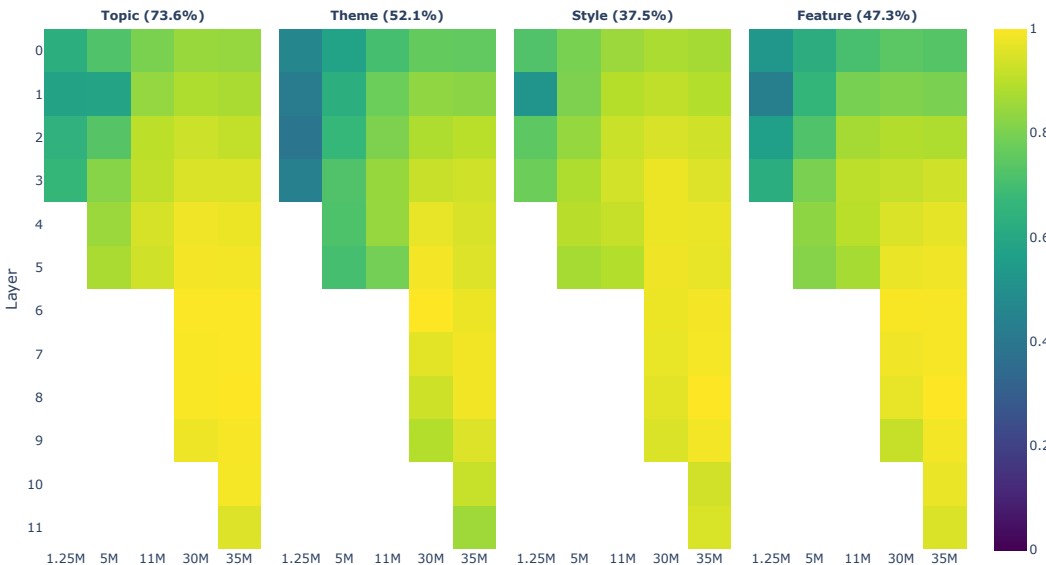

Figure 6: The relative probe accuracies for four story labels across model sizes and layers. The maximal absolute score is shown at the top. Probe accuracy is consistently best for the largest model (30M) at layer 6. The smallest model (1.25M) is often only able to achieve 50% relative accuracy.

on TinyStories and SimpleStories. These models use bilinear MLPs, which are easier to interpret in an input-independent fashion [Pearce et al., 2024, Elhage et al., 2021]. Specifically, each output token of these simple models is computed through a matrix of token pairs, encoding (skip-)trigrams of the form [input, input → output] such as ['three', 'little' → 'pigs']. The current tokens and attention layer determine these interactions. If we set aside attention, the current token can only interact with itself toward an output, yielding a bigram matrix. Importantly, this bigram matrix contains information about the MLP rather than simply the embedding and unembedding [Pearce et al., 2024]. While this simplification doesn't include cross-token mechanisms, this weight-based technique provides a straightforward picture of what is important to these simple models.

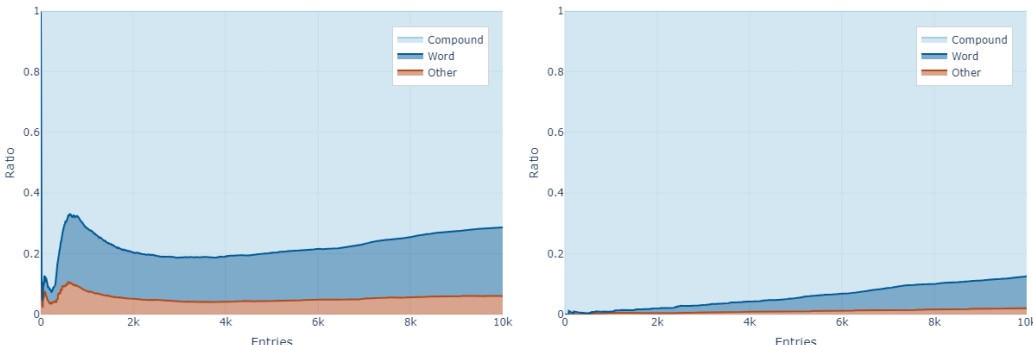

Figure 7: Cumulative categorization of the 10k highest outliers in bigram tables for a TinyStories (left) and SimpleStories (right) model. The word bigram distribution differs strongly, plausibly due to their overrepresentation in TinyStories (Table 1) .

This bigram matrix [input → output] contains about 10M entries, indicating the importance of any input token toward any output token. The entries' magnitudes are normally distributed with a long tail of outliers, likely indicating noteworthy patterns in the model. We study these outliers and organize them into three categories: *compound*, *word* and *other*. Compound bigrams result from the tokenizer splitting words into pieces such as ['decor', 'ations']. Unsurprisingly, the compound bigrams are highly prevalent in the outliers for both models (Figure 7). This is where the similarities end; the TinyStories model learns more word-based bigrams (i.e. two adjacent words), likely due to some frequently occurring n-grams (shown in Table 1). Manual analysis corroborates this: the bigram [bos, 'once'] is the strongest outlier, and ['once', 'upon'] appears at rank 70 along with many others in the top 1,000. This indicates that dataset diversity has a tangible impact on model weights. Lastly, we also find traces of rogue (non-ASCII) characters in TinyStories models, making up roughly 6-10% of outliers. These are aberrations of the dataset [3] and are strongly reflected in the weights. In contrast, the SimpleStories word bigrams are more uniformly distributed across outliers (Figure 7).

The models studied in this section have 8M parameters (more than half of which are the embeddings), using $d_{model} = 512$ and $n_{heads} = 8$. They are trained for a single epoch and qualitatively produce somewhat coherent stories.

## 4 Related Work

Apart from the seminal work of Eldan and Li [Eldan and Li, 2023] which we have discussed at length here, there have been other advances in the tiny model domain. Our efforts to provide a fully open-source LLM training process have been inspired by the pioneering OLMo [Groeneveld et al., 2024, OLMo et al., 2024] model family. Recent efforts to enter into competition with much larger models on benchmarks such as HellaSwag [Zellers et al., 2019] and BLiMP [Warstadt et al., 2020] have been undertaken in [Hillier et al., 2024], boasting models of size 10M-100M and containing architectural innovations. Similarly, the BabyLM challenge [Warstadt et al., 2025] invited submissions of trained model architectures optimizing the training sample efficiency on a given natural language corpus, and received many submissions with 10M-100M parameters.

## 5 Limitations

Because of our focus on the training data and tokenization, we do not examine how model architectures affect the output quality. Additionally, while we demonstrate the benefits of custom tokenization across different datasets and compare various tokenization and architectural strategies, we did not explore other tokenization approaches such as BPE tokenizers [Sennrich et al., 2015, Gage, 1994] to provide a more comprehensive comparison with our semantically informed tokenizer. We also did not examine how different model architectures beyond the Llama variant might interact with our dataset and tokenization improvements.

---

[3]Some TinyStories contain strings of non-ASCII characters, possibly due to some encoding error.

Having trained and evaluated models with as few as 1.25M parameters, a natural extension would be to find the lower limit for a model that still outputs grammatical language, given our improved sample efficiency. We have not done so here.

While our explorations with bilinear MLPs and linear probes aim to demonstrate how our dataset can aid interpretability research, there is currently no good substitute for the subjective judgment of ease of use by researchers. As we saw in our analysis, the diversity of text datasets is easier to operationalize. However, diversity has many facets, and automated metrics are imperfect, so we also recommend manually comparing random samples from our datasets with alternatives to get a grounded picture.

Finally, some parts of our evaluation exclusively treats our English dataset, due to the lack of a fair comparison to, say, the reading grade. For other languages, it might be necessary to consider different aspects for a comprehensive assessment.

## 6 Outlook

Based on the above, we recommend our English SimpleStories dataset over TinyStories for training and analyzing small language models. SimpleStories presents a more challenging language modeling problem that includes more diverse grammatical and syntactic patterns, while staying firmly in the realm of simple language and fiction. As the costs of sampling from openly available models continue to decrease, we anticipate that it will become more common to create similar synthetic datasets to ours for language modeling research. We invite the community to do so by providing a dataset creation repository, at https://github.com/simple-stories/simple_stories_generate. A comparative analysis of such datasets may follow the methods presented in this work.

**Acknowledgements**

We thank Nix Goldowsky-Dill, Rylan Schaeffer, Joseph Bloom, Joseph Miller and Alice Rigg for valuable feedback and insight into the wanted specifications for this dataset. This research was funded by Dan Braun and Anthropic. This research received funding from the Flemish Government under the "Onderzoeksprogramma Artificiële Intelligentie (AI) Vlaanderen" programme.

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

# A  Story Generation

Below is the story generation prompt for SimpleStories, with values of changing parameters in {curly brackets}. A grammar feature is used in 50% of stories, an author persona is specified in 33% of samples. Paragraph counts are uniformly distributed between 1 and 9, the number of stories in one completion is inversely proportional to this.

> Write {12} short stories ({2} paragraphs each) using very basic words. Do not number each story or write a headline. Make the stories diverse by fully exploring the theme, but each story should be self-contained. Separate the stories by putting {"The End."} in between. Make the stories as qualitatively distinct to each other as possible. In particular, never start two stories the same way! Each story should be about {Responsibility}, include {secret societies}, be {lyric} in its writing style and ideally feature {inner monologue}. The most important thing is to write an engaging easy story, but where it makes sense, demonstrate the use of {progressive aspect}. Write from the perspective of {someone curious}. If you need to use proper names, make them from space-separated common words. Either don't give characters a name, or select from {list of names}. Complex story structure is great, but please remember to only use very simple words! If you can, start the story with {a noun} that begins with the letter {p}.

The parameters stem from the following set of options.

> **Theme**: Friendship, Courage, Contradiction, Coming of age, Kindness, Amnesia, Adventure, Imagination, Family, Perseverance, Curiosity, Honesty, Romance, Teamwork, Responsibility, Strategy, Magic, Discovery, Betrayal, Deception, Generosity, Creativity, Self-Acceptance, Helping Others, Hardship, Agency, Power, Revenge, Independence, Problem-Solving, Resourcefulness, Long-Term Thinking, Optimism, Humor, Love, The Five Senses, Tradition, Innovation, Hope, Dreams, Belonging, Travel, Overcoming, Trust, Morality, Happiness, Consciousness, Failure, Conflict, Cooperation, Growth, Loss, Celebration, Transformation, Scheming, Challenge, Planning, Wonder, Surprises, Conscience, Intelligence, Logic, Resilience.
>
> **Topic**: talking animals, fantasy worlds, time travel, a deadline or time limit, space exploration, mystical creatures, underwater adventures, dinosaurs, pirates, superheroes, fairy tales, outer space, hidden treasures, magical lands, enchanted forests, secret societies, robots and technology, sports, school life, holidays, cultural traditions, magical objects, lost civilizations, subterranean worlds, bygone eras, invisibility, giant creatures, miniature worlds, alien encounters, haunted places, shape-shifting, island adventures, unusual vehicles, undercover missions, dream worlds, virtual worlds, riddles, sibling rivalry, treasure hunts, snowy adventures, seasonal changes, mysterious maps, royal kingdoms, living objects, gardens, lost cities, the arts, the sky
>
> **Style**: whimsical, playful, epic, fairy tale-like, modern, classic, lyric, mythological, lighthearted, adventurous, heartwarming, humorous, mystical, action-packed, fable-like, surreal, philosophical, melancholic, noir, romantic, tragic, minimalist, suspenseful
>
> **Narrative Feature**: dialogue, in medias res, a moral lesson, absence indicating a presence, a story told through letters, a twist ending, an unreliable narrator, foreshadowing, irony, inner monologue, symbolism, a MacGuffin, a non-linear timeline, a reverse timeline, circular narrative structure, a flashback, a nested structure, a story within a story, a Red Herring, multiple perspectives, Checkhov's gun, the fourth wall, a cliffhanger, an anti-hero, juxtaposition, climactic structure

**Grammar Feature**: present tense, past tense, future tense, progressive aspect, perfect aspect, passive voice, conditional mood, imperative mood, indicative mood, relative clauses, prepositional phrases, indirect speech, exclamatory sentences, comparative forms, superlative forms, subordinate clauses, ellipsis, anaphora, cataphora, wh-questions, yes-no questions, gerunds, participle phrases, inverted sentences, non-finite clauses, determiners, quantifiers, adjective order, parallel structure, discourse markers, appositive phrases

**Author Persona**: an explorer archetype, a rebellious author, a powerful leader, a wise, old person who wants to teach the young, an innocent author, a moralistic teacher, a hopeless romantic, a hurt, ill-intentioned person, an academic, a jester archetype, a poet, a philosopher, a mother, a father, someone curious, someone evil, someone who wants to prove a point, a child, a pedant, the everyman, the oppressed, a cruel person, someone who loves order and structure

## B  Model-as-a-Judge Evaluation

The following prompt was used to evaluate the story datasets, given sets of four stories each.

Please evaluate this set of stories and provide structured feedback.

Stories to evaluate: {stories_string} Analyze these stories and provide scores (0-100) and brief explanations for: 1. Simplicity: How easy are the stories to understand? 2. Diversity of style: How varied is the writing style across stories? 3. Diversity of content: How varied are the themes and plot lines?

Provide your assessment in this exact format, for all stories taken together: {{"explanation": "short explanation here", "simplicity": 0, "diversity_style": 0, "diversity_content": 0}}

The following prompt was used to evaluate the completions by all mentioned tiny models.

Evaluate the following story based on four criteria by assigning each a score from 0 to 100: 1. **Originality**: Rate the creativity and uniqueness of the story. 2. **Coherence**: Rate the logical flow and consistency of the story. 3. **Grammar**: Rate the grammatical correctness of the story. Ignore spacing and capitalization. 4. **Quality**: Rate the overall quality of the story. You should also provide a short explanation for your judgment.

**Story to evaluate:** {story}

Please provide your assessment in the following format, ensuring each score is an integer between 0 and 100: {{"EXPLANATION": "The dialogue is coherent, but the phrasing is slightly off.","ORIGINALITY": 0, "COHERENCE": 0, "GRAMMAR": 0, "QUALITY": 0}}

## C  Probe setup

Our probes classify the intermediate activations for all tokens ($a_t$) in a story. Instead of learning a massive matrix $\mathbb{R}^{D \times T} \to \mathbb{R}^L$, we first pool all activations per story, followed by a linear head.

$$
\begin{aligned}
\textbf{Average pooling:} \quad & w_t = 1/|T| \\
\textbf{Weighted pooling:} \quad & w_t = a_t p/|T| \\
\textbf{Softmax pooling:} \quad & w_t = \exp(a_t p)/ \textstyle\sum_{i=1}^{T} \exp(a_i p)
\end{aligned}
$$

Here, $w_t$ represents the weight of each token before being summed and $p \in \mathbb{R}^T$ represents a learned vector. We found that average pooling performed subpar, probably unable to filter out uninformative token representations. On the other hand, we found softmax pooling to be overly specific, often fixating on only a single token. The ordinary weighted pool performed best by far.

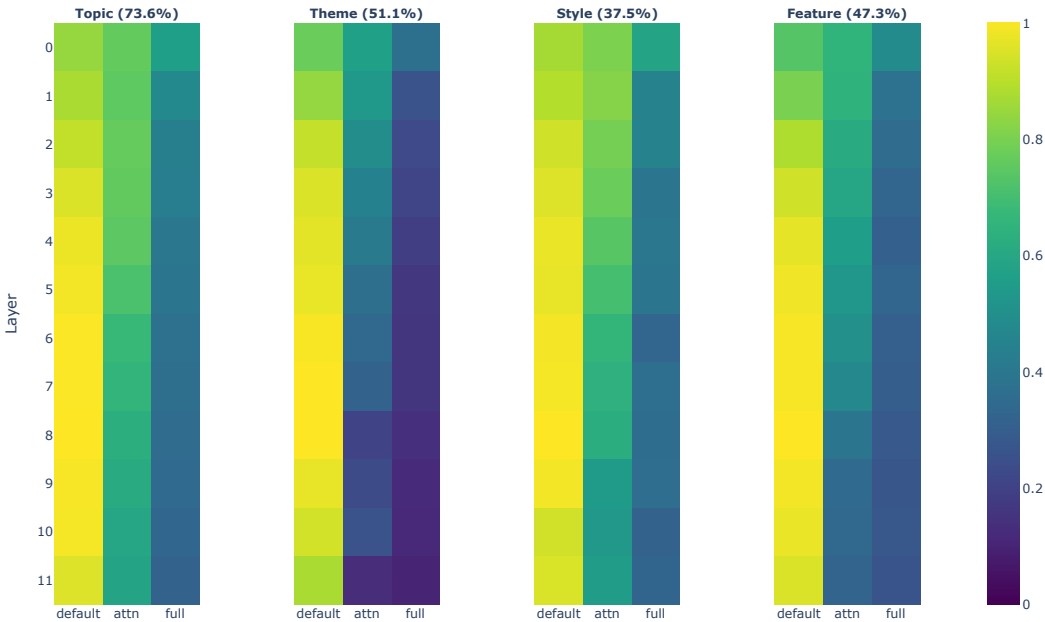

Figure 8: Sanity checks for probe accuracy using the learned 35M model (default), the same model with ablated attention (attn) and a fully re-initialized model (full). The ablated attention probes retain 50%-80% of relative accuracy, while the fully re-initialized models only achieve about 40%. This indicates the trained model does learn representations that are useful for the target labels.

Across experiments, we trained the probes over $2^{18}$ stories (roughly 250k stories). Activations were sampled from the residual stream at the layer output. All probes use the Muon optimizer [Jordan et al., 2024] with a learning rate of 0.05. Different setups generally attain equal accuracies but require longer training times.

To test whether the trained probes yield false positives (finding spurious correlations), we perform the same analysis on (partially) re-initialised models [Adebayo et al., 2020]. This sanity check determines the effect of learned model parameters on the probe accuracy versus random spurious correlations from the inputs. This reveals that, while probes on randomized models still achieve better-than-chance accuracy, the probes on the fully trained network remain far superior (see Figure 8). Lastly, we verify how robustly these representations are encoded through fuzzing – adding noise to the activations during probe training. Across experiments, moderate noise only impacted accuracy by a few per cent, indicating the learned probes are unlikely to be spurious.

## D   Bigram tables

Some of the most common bigrams are shown in Table 3 for TinyStories and Table 4 for SimpleStories.

## E   Syntactic Templates

Syntactic templates [Shaib et al., 2024], i.e., the most frequent part-of-speech (POS) n-grams, are shown in Table 5 for both SimpleStories and TinyStories (n=6), along with the fraction of stories containing at least one such template.

We note the POS sequences appearing in SimpleStories are move evenly distributed (mostly between 10.7% and 19.2%).

Table 3: Cherry-picked (but representative) samples from the top 100 (out of 10M) entries from the TinyStories bigram matrix. About 10% are actual word bigrams (like 'ran to'), about 80% are compound bigrams (indicated by ##), and 10% are rogue tokens (such as œand €).

| Rank | Input | Output |
|------|-------|--------|
| 0 | [BOS] | once |
| 1 | prov | ##ided |
| 6 | ran | to |
| 14 | yaw | ##ned |
| 27 | œ | ##g |
| 29 | creat | ##ions |
| 44 | couldn | ' |
| 54 | complet | ##ing |
| 71 | € | " |
| 94 | decided | to |

Table 4: Cherry-picked (but representative) samples from the top 100 (out of 10M) entries from the SimpleStories bigram matrix. All except one ('want to') are compound bigrams instead of word bigrams.

| Rank | Input | Output |
|------|-------|--------|
| 0 | anx | ##iety |
| 1 | ripp | ##led |
| 4 | cur | ##led |
| 10 | ripp | ##les |
| 12 | complet | ##ely |
| 20 | sli | ##ding |
| 33 | spir | ##al |
| 34 | emot | ##ions |
| 58 | pumpk | ##ins |
| 84 | want | to |

Table 5: Syntactic templates [Shaib et al., 2024] across SimpleStories and TinyStories (n=6) and their frequency in each dataset. Examples are sourced from TinyStories in those cases where they exist in TinyStories (otherwise, they are sourced from SimpleStories).

| POS Sequence | SimpleStories | TinyStories | Example |
|---|---|---|---|
| DT JJ NN VBN NNP . | 0.0 | 100.0 | a little girl named Lily. |
| , EX VBD DT JJ NN | 0.0 | 34.0 | , there was a big cat |
| VBD DT JJ NN VBN NNP | 0.0 | 29.9 | was a little boy named Mark |
| EX VBD DT JJ NN VBN | 0.0 | 29.5 | there was a silly cat named |
| VBD IN DT NN CC VBD | 0.0 | 14.1 | ran to the bathroom and saw |
| RB IN DT NN EX VBD | 0.0 | 13.0 | Once upon a time there lived |
| VBD TO VB IN DT NN | 0.0 | 12.8 | wanted to play with the puppy |
| IN DT NN EX VBD DT | 0.0 | 12.0 | In the attic there was a |
| PRP VBD IN DT NN CC | 0.0 | 10.2 | He ran to the door and |
| IN DT NN IN PRP$ NN | 0.0 | 8.9 | to the bathroom with his mom |
| VBD RB JJ IN PRP VBD | 0.0 | 8.9 | was so excited that she ran |
| DT NN EX VBD DT JJ | 0.0 | 8.1 | a time there lived a big |
| PRP VBD TO VB IN DT | 0.0 | 8.0 | he decided to hide in the |
| PRP VBD DT NN CC VBD | 0.0 | 7.8 | She grabbed a napkin and wiped |
| DT JJ NN IN DT NN | 16.6 | 15.1 | a small bird on the grass |
| PRP VBD DT JJ NN IN | 14.3 | 13.0 | she saw a big armchair near |
| VBD DT NN IN DT NN | 18.2 | 12.8 | chased the ball down the hill |
| IN DT NN IN DT NN | 19.2 | 11.6 | in a bush near the puddle |
| VBD DT JJ NN IN DT | 12.3 | 11.6 | saw a delicate leaf on the |
| PRP VBD TO VB DT NN | 13.8 | 11.3 | He wanted to see the wolf |
| , PRP VBD DT JJ NN | 28.4 | 11.2 | , she felt a strange tingling |
| PRP VBD DT NN IN DT | 15.7 | 9.0 | she put the boat in the |
| , DT NN VBN NNP VBD | 13.9 | 0.0 | , a dinosaur named Lily spotted |
| DT NN VBD IN DT NN | 18.4 | 0.0 | The beast thought for a moment |
| DT NN VBD DT NN IN | 12.6 | 0.0 | the moonlight bathed the car in |
| , PRP VBD IN DT NN | 14.9 | 0.0 | , he wept for the world |
| VBD DT NN IN PRP$ NN | 12.7 | 0.0 | felt the wind on her face |
| PRP VBD DT NN IN NN | 13.0 | 0.0 | they filled the kingdom with love |
| PRP VBD IN DT JJ NN | 15.9 | 0.0 | he was in a magical land |
| , DT NN VBD IN DT | 10.8 | 0.0 | , a man stared at an |
| DT NN IN DT JJ NN | 15.9 | 0.0 | a story about a clever rabbit |
| , PRP VBD DT NN IN | 18.4 | 0.0 | , she dipped the twig into |
| DT NN VBD DT JJ NN | 11.3 | 0.0 | the boy felt a deep emptiness |
| NN VBD IN DT JJ NN | 10.7 | 0.0 | sky turned to a deep blue |

Table 6: As in Table 1, top 20 most frequent 5-grams in our Japanese dataset, where tokens were computed by the MeCab tokenizer [McCann, 2020], for our Japanese tokenizer, filtered for overlaps of more than two tokens.

**SimpleStories-JP**

| Frequency | Phrase | English Translation |
|---|---|---|
| 14.69% | こと・に・し・まし・た | decided to |
| 8.15% | い・まし・た・彼・は | [past tense] he [topic] |
| 7.99% | い・まし・た・ある・日 | [past tense] one day |
| 7.01% | 遊ん・で・い・まし・た | played |
| 6.41% | し・て・い・まし・た | did |
| 6.39% | 住ん・で・い・まし・た | lived |
| 6.24% | い・まし・た・彼女・は | [past tense] she [topic] |
| 5.92% | て・い・まし・た・ある | [past tense] one/a |
| 5.74% | 時間・を・過ごし・まし・た | spent time |
| 5.42% | が・住ん・で・い・まし | [subject] lived |
| 5.39% | まし・た・彼・ら・は | [past tense] they [topic] |
| 5.22% | 楽しい・時間・を・過ごし・まし | had a good time |
| 5.15% | こと・が・でき・まし・た | accomplished |
| 5.01% | が・い・まし・た・彼 | [subject] was. He |
| 4.75% | まし・た・そう・すけ・は | [past tense] Sousuke [topic] |
| 4.54% | まし・た・そこ・に・は | [past tense] there [topic] |
| 4.41% | は・友達・と・一緒・に | [topic] together with friends |
| 3.68% | よ・と・言い・まし・た | said |
| 3.62% | て・い・まし・た・彼 | [past tense]. He |
| 3.59% | で・遊ん・で・い・まし | [location] played |

