# OpenReview forum: "Parameterized Synthetic Text Generation with SimpleStories"
_NeurIPS.cc/2025/Datasets_and_Benchmarks_Track — NeurIPS 2025 Datasets and Benchmarks Track poster_

### Official Review · Reviewer_zAVm · 2025-07-01

**Ethics Flags:** Discrimination, bias, and fairness
**Rating:** 5
**Confidence:** 5

**Summary:**

This paper presents an open-source corpus of 2M synthetic children's stories (in both English/Japanese) called SimpleStories. These stories are generated with parameterized and prompt-controlled features, such as topic, theme, style, and narratives. Compared to previous datasets like TinyStories, SimpleStories offers much greater linguistic diversity and richer labeling. The authors also release a suite of small language models trained on this data, which outperform similar-sized models on originality, coherence, and grammar, making SimpleStories a valuable resource for interpretability and language modeling research.

**Dataset Code Accessibility:**

Partly

**Dataset Code Comments:**

The final dataset is hosted on Huggingface, but w/o the original generation script.

The code for reproducing the learnability and data quality analysis is hosted on Github and accessible.

**Ethical Comments:**

As is pointed out in the weakness section, given that the label-controlling during the generation is far from being perfect, it is likely that the distribution of the generated data is impacted by the existing bias and preferences of the LLMs the authors use to craft this dataset. If this should be the case, using this dataset to train new models could amplify these biases.

**Ethical Considerations:**

No, there are no or only very minor ethics concerns

**Final Justification:**

The authors provide legit improvements in evaluation experiments and promises to update manuscript in a better logic with more carefully conducted ablation. I'd like to raise my score to reflect this improvement of soundness.

**Limitations Weaknesses:**

- According to the original report by the authors, it is concerning that the controlling labels are not very perfectly ensured (25%-50%, still significantly better than random) by the final output. It raises such a problem that could weaken the core claims of the paper: if the diversity of the original intent features is good, but these intent are never sufficiently fulfilled, will the generated stories - thus the presented dataset - still be considered diverse in such attributes?

- It is arguable that the presented language model being the "smallest" model that produces grammatical English can all be due to the dataset quality. Given that the presented language models is trained using a much smaller vocabulary size (which in my opinion naturally make it even easier to produce less noisy outputs), it's probably a fairer comparison to also report TinyStories33M w/4096SimpleTokenizer.

**Strengths Contributions:**

- Compared to its forefathers like TinyStories, the diversity of attributes and language usage of SimpleStories is much better.
- Generated stories are also companioned by their original prompted labels, easier to faciliatate controllable generation study.
- Good learnability study using different model sizes is conducted, showing how effectively this dataset can be compressed and learnt.

---

> ### Author Rebuttal · Authors · 2025-07-31
>
> Many thanks for the insightful review! Regarding strengths, the reviewer states that we have successfully demonstrated semantic diversity of our dataset, and that the labels will be useful for future work. We address the raised limitations:
>
> **Limitation 1**
> > According to the original report by the authors, it is concerning that the controlling labels are not very perfectly ensured (25%-50%, still significantly better than random) by the final output. It raises such a problem that could weaken the core claims of the paper: if the diversity of the original intent features is good, but these intent are never sufficiently fulfilled, will the generated stories - thus the presented dataset - still be considered diverse in such attributes?
>
> This is a valid point about an essential argument in the paper, so we decided to conduct a new experiment that better isolates label quality. To address the fact that the experiment the reviewer is referencing uses (1) an off-the-shelf weak model (gpt-4o-mini) and (2) no calibration or normalization of the outputs, we run the following setup: We use a stronger model (o4-mini) to tell apart two pairs of stories A and B, where pair A contains two stories with the same label, and pair B contains two stories with different labels, in one given category (topic, theme, style, feature). We also provide the values of the categories that are not tested and the three values of the tested label category that occur across pairs A and B in the prompt. This design should remove biases towards a given value of a label. Using this, again with N=200 for each label category, we get classification accuracies of 89.5%, 77.5%, 73%, 74.5% for topic, theme, style, and feature, for the English dataset. Random chance would give 50%, and we can reject the null hypothesis that the accuracies are no better than chance with p-values smaller than 10^-43, 10^-16, 10^-11, 10^-13, respectively. This confirms our suspicion that the previous experiment relied too heavily on the (biased) judge model. The previous experiment was merely a lower bound, whereas here we show that that the labels are higher quality than we thought, by isolating them from the other label categories separately. (To illustrate the previous failure mode: A story with the theme “Adventure” that is also written in a “melancholic” style will seem more like it is about the theme “Loss” simply because melancholy and themes of loss are correlated.)
> As per this year's regulations we cannot show the updated manuscript, but of course these results are included in the final version.
>
> [Less important: There is also more context on the label accuracy in Section 3.6. There, we train and evaluate linear probes on our language models to see if their activations on in-distribution samples can predict the labels of the samples. For all categories of samples (topic, theme, style, feature), the label prediction accuracies were much higher than when asking a judge model. (For sanity checks etc., see Appendix C.)
> The sample categories must consequently be encoded in the language model’s weights, which in turn means that the samples with different labels must also be distinct enough to allow this.
> Finally, we want to stress that we are sure to advance the state of the art in semantic diversity, as the comparison with TinyStories in Section 3.2 and Figure 1 (Right) shows. We are therefore confident that our paper constitutes a significant step forward in this area.]
>
> **Limitation 2**
> > It is arguable that the presented language model being the "smallest" model that produces grammatical English can all be due to the dataset quality. Given that the presented language models are trained using a much smaller vocabulary size (which, in my opinion, naturally makes it even easier to produce less noisy outputs), it's probably a fairer comparison to also report TinyStories33M w/4096SimpleTokenizer.
>
> We decided to newly conduct this experiment also, comparing to the TinyStories 33M model with Llama architecture and a vocabulary size of 4096: TinyStories-33M of Llama architecture with our custom tokenizer achieved scores of 43/77.2/80.9/61.3 (originality/coherence/grammar/quality), showing modest improvements over the original TinyStories-33M Llama model in coherence (+8.2) and grammar (+25.9). However, the performance gap between TinyStories and SimpleStories datasets remains substantial regardless of tokenizer choice, with our SimpleStories-35M models still significantly outperforming this improved baseline across all metrics. This confirms that while tokenizer design provides some benefit, dataset quality is the primary driver of our performance improvements, validating our core contribution.
>
> See also the new experiment in response to review 4huu; it tests for something similar.
>
> **Regarding the Dataset Code Comment**
>
> > The final dataset is hosted on Huggingface, but w/o the original generation script.
>
> The original generation script is available at the repository cited in line 60 on page 2, in the script generate_stories.py for unbatched API calls and oai_batch.ipynb for batched API calls. We changed the wording to be more clear that the generation script is accessible there.
>
> We thank you again for your comments. If we have adequately addressed your concerns, we would kindly ask you to reassess your review score, taking this rebuttal into account.

---

> > ### Comment · Reviewer_zAVm · 2025-08-01
> > **Re Rebuttal**
> >
> > I'm fully convinced by the authors. Should the promised updates be fulfilled in the final manuscript, I'd like to further endorse this paper for acceptance.

---

> > > ### Author Response · Authors · 2025-08-04
> > >
> > > Many thanks for the endorsement!

---

### Official Review · Reviewer_19o5 · 2025-07-02

**Rating:** 4
**Confidence:** 3

**Summary:**

This paper presents SimpleStories , a diverse, labeled synthetic story dataset for training small language models. Compared to TinyStories, it improves interpretability, diversity, and efficiency. Models trained on SimpleStories outperform baselines across multiple metrics

**Dataset Code Accessibility:**

Yes

**Ethical Considerations:**

No, there are no or only very minor ethics concerns

**Limitations Weaknesses:**

- The study focuses primarily on data and tokenizer improvements without systematically exploring alternative or modern architectures (e.g., Diffusion Language Models).
- Although the dataset includes both English and Japanese stories, all evaluations—including diversity metrics, labeling accuracy, and model performance are conducted only on the English subset.
- Dataset link and code link should be provided in the first page, given the fact that this is dataset benchmark paper.
- Some figures are really blurry, please polish it, it's low-quality.

**Strengths Contributions:**

- The entire dataset creation process and training code are fully open-sourced, making it easy for others to build on the work. Plus, the dataset isn’t limited to English—Japanese stories are included too, which opens the door for multilingual research.
- Supports Interpretability: The dataset is thoughtfully designed to support interpretability studies. It includes labeled features like theme and style, and the models are compatible with tools like bilinear MLPs and linear probes to better understand what the model has learned.

---

> ### Author Rebuttal · Authors · 2025-07-31
>
> Thanks for the review! As described below, this led us to improve the manuscript in several different ways.
>
> As for strengths, the reviewer highlights that we not only study the dataset via classic NLP techniques, but also via various kinds of learnability by models.
>
> **Limitation 1**
> > The study focuses primarily on data and tokenizer improvements without systematically exploring alternative or modern architectures (e.g., Diffusion Language Models).
>
> This is correct in spirit; our experiments do not compare other architectures with the standard self-attention decoder-only architecture for e.g. output quality. After having discussed training an RNN-based model as well, we opted to focus our experiments on the dataset itself.
>
> That said, we are definitely excited by new architectures, particularly those that aim to increase the interpretability of the model. We would like to direct the reviewer’s attention to Section 3.7, where we explore a recent architectural innovation, namely bilinear MLPs. We trained a bilinear MLP language model on both our dataset and the comparison. This demonstrated that our dataset shows fewer spurious correlations. We also believe that novel architectures should be very useful as tools to understand the training distribution, and have hopefully made it easy for people to build on our work to explore this.
>
> **Limitation 2**
> > Although the dataset includes both English and Japanese stories, all evaluations—including diversity metrics, labeling accuracy, and model performance are conducted only on the English subset.
>
> This was a significant limitation of our initial submission and thank the reviewer for bringing this issue up! We have taken this comment as an opportunity to run experiments on the Japanese dataset as well. This year’s guidelines prevent us from sharing a manuscript revision. Still, we gladly share the results in text form: For the model-judged simplicity and content diversity, we observe similar values for the Japanese dataset as for the English dataset, with much higher diversity and similar simplicity as compared to TinyStories. Diversity in style and content was slightly lower than in the English dataset, while simplicity was within error bars. In numbers: Simplicity, content diversity and style diversity are at 84.9, 73.1, 61.8 for Japanese as compared to 84.8, 61.3, 47.1 for TinyStories and 84.7, 75.7, 63.5 for our English dataset. The label recovery accuracy is similarly good or better than for the English version for the attribute “theme”. In particular, “topic” and “theme” are very clearly recovered much better than chance. Label recovery accuracies for “topic”, “feature” and “style” are lower than for the English dataset. In numbers, we get 42%, 23%, 8.5%, 14% for topic, theme, style and feature as compared to 49%, 22.5%, 14.5%, 15.5% for English.
> As for the n-gram frequency analysis, we newly include a table with 5-grams, meaning sequences of 5 tokens as computed by the MeCab Japanese tokenizer. Here, as with the n-gram table for our English dataset, we qualitatively don’t see the irregularities present in TinyStories. Of course we are very open to hearing more suggestions for experiments.
>
> [Broader remarks: Some difficulty comes from the circumstance that no other synthetic dataset in Japanese is available for a fair comparison, thus turning any benchmarks into more of a summary statistic than a legible metric of quality. Further, some metrics we used on the English dataset, such as the Flesh-Kincaid reading score, simply don’t admit an analogue, so the analysis above is only the subset of experiments that still apply.]
>
>  **Limitation 3**
> > Dataset link and code link should be provided in the first page, given the fact that this is dataset benchmark paper.
>
> We have now put the dataset link and code on the first page. Usability feedback is very valuable to us, thanks!
>
> **Limitation 4**
> > Some figures are really blurry, please polish it, it's low-quality.
>
> We have replaced all figures with crisp versions.
>
> ---
>
> We thank the reviewer again for their helpful comments and questions! We kindly ask that they reassess the review score if their concerns were adequately addressed by this response, and by the improvements we could make as a result of the review.

---

> > ### Comment · Reviewer_19o5 · 2025-08-07
> >
> > Thanks for your effort, my initial concerns are fully addressed. I’m happy to maintain my current score and support acceptance of the paper.

---

> ### Author Response · Authors · 2025-08-06
>
> We are taking yesterday's discussion period reminder as an opportunity to ping about the response to your review. We are curious to know if our adjustments addressed your concerns, and greatly look forward to your reply.

---

### Official Review · Reviewer_4huu · 2025-07-03

**Rating:** 5
**Confidence:** 5

**Summary:**

This paper presents SimpleStories, a large-scale open-source synthetic story dataset with 2 million English and 2 million Japanese samples, designed for training and analyzing small language models.

**Dataset Code Accessibility:**

Yes

**Ethical Considerations:**

No, there are no or only very minor ethics concerns

**Final Justification:**

Thanks for the detailed rebuttal and additional experiments. While some of my original concerns—particularly about interpretability with small models and real-world applicability—remain, I appreciate the clarifications and the extra effort put into addressing them. I understand the practical constraints and, while not fully convinced, I find the response reasonable enough to accept.

**Limitations Weaknesses:**

1.	The tokenizer comparison is a bit unfair since TinyStories didn’t get its own optimized tokenizer.
2.	The interpretability analysis only uses very small models, which limits how far the insights can go.
3.	Since it’s all synthetic text, we don’t know how well it generalizes to real-world tasks like NLU or generation.

**Strengths Contributions:**

1.  The authors improve upon TinyStories by addressing key limitations like lack of labeling, poor diversity, and non-open release. TinyStories is a representative interpretable small LLMs dataset.
2.  They provide a cleaner, more controllable dataset that avoids repetition, filters harmful content, and adds rich annotations.
3.  The dataset supports multilingual generation (English and Japanese), which TinyStories did not.
4. The work fills a gap by offering a labeled, diverse dataset ideal for probing and analyzing small language models.
5.  It also enables new research into training efficiency and the minimal scale needed for grammatical language generation.

---

> ### Author Rebuttal · Authors · 2025-07-31
>
> We are grateful for the reviewer’s time and effort. We are happy to learn that the reviewer believes that our work unlocks new lines of research in future work, and that we could adequately demonstrate the quality of the annotations and dataset diversity. Regarding the limitations, we respond point-by-point:
>
> **Limitation 1**
> > The tokenizer comparison is a bit unfair since TinyStories didn’t get its own optimized tokenizer.
>
> We conduct an additional experiment training TinyStories-33M with a custom tokenizer optimized for the dataset, which achieved scores of 41.0 (originality), 77.0 (coherence), 82.6 (grammar), and 60.8 (quality) - representing improvements of +1.0, +31.0, -2.4, and +10.8 points respectively over the baseline TinyStories with GPT-2 tokenizer, yet still remaining substantially below our SimpleStories-35M model across all metrics. This result confirms that while tokenizer optimization provides some benefits (particularly for coherence), the substantial performance gains of our approach are primarily attributable to our improved dataset rather than tokenizer or other technical advantages.
> [As per this year's regulations, we cannot show the updated manuscript during the discussion phase, but naturally these results are for inclusion in the final version main text, or appendix if space does not suffice.]
>
> **Limitation 2**
> > The interpretability analysis only uses very small models, which limits how far the insights can go.
>
> The interpretability section is intended to verify whether the dataset improvements are reflected in model internals. Current methods don’t admit precise statements about deep models, so we use multiple methods, applying large-scale probing along with low-level experiments on a smaller model. The former shows that all our models learn high-level attributes we care about, while the latter shows how the low-level learned structures (bigrams) differ. If the reviewer has any suggestions in mind, we would love to hear them.
>
> **Limitation 3**
> > Since it’s synthetic, we don’t know how well it generalizes to real-world tasks.
>
> We assume the reviewer means either our model suite or our dataset. We respond to both.
> Tiny models trained on synthetic data, such as ours, are indeed mostly applicable as toy models of production-grade language models. They are not expected to be practically helpful for real-world tasks, as even our largest models have orders of magnitude fewer parameters than frontier models. Luckily, we can already observe many works deriving value from tiny language models, such as in developing new techniques and insights in interpretability, where the low parameter count greatly improves research iteration speed. The methods developed using tiny models can then be applied to larger models on real-world tasks.
>
> We believe that synthetic data is better suited to the tiny model paradigm, because the language modelling problem they are solving is “nicer” than real-world corpora, which by their nature are often messy and too complex to be modelled by small models. Intuitively, we hope this dataset will be an ‘MNIST for language’, which is not applicable by itself but is undeniably valuable. Therefore, our contribution of an idealized corpus is useful precisely because it does not contain any information on real-world phenomena.

---

> ### Author Response · Authors · 2025-08-06
>
> Many thanks again for the review! As the discussion period approaches its end, we are wondering if our adjustments solved the limitations you pointed out, and whether you have further comments about the paper.

---

### Official Review · Reviewer_fjxx · 2025-07-04

**Rating:** 4
**Confidence:** 1

**Summary:**

SimpleStories offers a valuable resource for researchers in the field of NLP and synthetic data generation. The dataset's ability to induce syntactic and semantic diversity, combined with its focus on sample efficiency and model interpretability, makes it a powerful tool for advancing the development of language models. While the study has limitations, it provides a strong foundation for future research.

**Dataset Code Accessibility:**

Yes

**Ethical Considerations:**

No, there are no or only very minor ethics concerns

**Final Justification:**

As the authors response has relieved my concerns, so I am happy to raise the score to borderline accept.

**Limitations Weaknesses:**

1. the token size of the data is 602 million, may still be limited to train good models
2. methodological contribution seems limited

**Strengths Contributions:**

1. SimpleStories, a new fully open-source synthetic dataset consisting of simple yet diverse language suitable for pretraining and interpretability
2. detailed analysis of the diversity of SimpleStories
3. a suite of high-quality language models trained on SimpleStories

---

> ### Author Rebuttal · Authors · 2025-07-31
>
> We thank the reviewer for their comments. We appreciate that they've given their review the lowest confidence score (1), as we respectfully agree it doesn't deeply engage with the paper in comparison to the other reviews.
>
> We acknowledge that the review nonetheless points out that our work is a valuable resource and a strong foundation for future research. Regarding the limitations, we respond point-by-point:
>
> **Limitation 1**
> > the token size of the data is 2 million, may be limited to train good models
>
> Our dataset contains much more than 2 million tokens: We provide more than 2.1 million stories in both English and Japanese. Our English dataset alone contains 452 million words, or 602 million GPT-2 tokens. We have included these summary statistics for clarity in the final version.
> We can reassure the reviewer that quality language models can be trained on our data. Quoting from the review, we have already provided “a suite of high-quality language models trained on SimpleStories” to demonstrate to the community that this not only can be accomplished but is also practical and straightforward.
>
> **Limitation 2**
> > methodological contribution seems limited
>
> The standard for novelty in methodological contribution at this conference is rightfully high. We have therefore applied techniques from diverse fields to conduct a systematic end-to-end comparison of our dataset to the SoTA alternative. For instance, we are not aware of other work evaluating the merit of a pretraining dataset by conducting interpretability on models trained thereon (see Sections 3.6, 3.7). We are also filling a gap in the literature at the intersection of traditional NLP methods and pretraining data evaluation (see Sections 3.1, 3.3). We believe that data quality assessment methods will become more and more important as training data is increasingly sourced from synthetic generators, and believe that the tiny model regime, and our paper in particular, is at the forefront of data quality assessment methods.
>
> ---
>
> We hope to have cleared up the concerns. We kindly ask the reviewer to consider the other reviewers' assessments and, if they have no further comments, increase their rating.

---

> ### Author Response · Authors · 2025-08-06
>
> As recommended in yesterday's discussion period reminder, we would like to ping you about the response to your review. We are very interested to hear whether your outlook has changed from your initial assessment, in view of the other reviewers' opinions and our responses.

---

### Note · Authors · 2025-08-13

We are very happy with the feedback we received from the reviewers, which was uniformly positive among the three reviewers who gave a high confidence and/or engaged in the discussion. The consensus seems to be that we made a useful contribution and used strong methodology to back our claims. We are further glad that the discussion with reviewers zAVm, 4huu and 19o5 led to improvements which both we and, it seems, the respective reviewers think further strengthened the paper in several significant ways.
Regarding the comments of Reviewer fjxx, we would have hoped to receive an answer during the discussion phase, even though we understand that the time commitment for peer review can be demanding. Given they indicated the lowest confidence score of 1 and their lack of engagement, we would therefore like to ask the area chair to base their decision solely on the three other reviews, which all endorse our paper for acceptance.
We once again thank the reviewers and the area chair for their time!

---

### Decision · Program_Chairs · 2025-09-18

**Decision:**

Accept (poster)

**Comment:**

This paper proposed simplestories dataset, an extension of tinyimage dataset with similar methodology by synthetic data generation. Simplestories can also provide labels, and improve on diversity. Reviewers acknowledge the open-source efforts on dataset and code, the extensive analysis of the dataset, and the extension to multilingual (both English and Japanese datasets). During rebuttal, authors provide additional evaluation on the Japanese dataset.
However, reviewers also raised concerns on the novelty of the data generation process, and the accuracy of the labels. In general, it would be nice to have simplestories presented at the conference to encourage new dataset as reviewers are in favor of acceptance. On the other hand, the data generation method is too simple as criticized by reviewers, and I would hope to see more evidences showing simplestories is a useful dataset by itself, beyond comparing to tinystories.